# NEURAL OBLIVIOUS DECISION ENSEMBLES FOR DEEP LEARNING ON TABULAR DATA

**Sergei Popov**
Yandex
sapopov@yandex-team.ru

**Stanislav Morozov**
Yandex
Lomonosov Moscow State University
stanis-morozov@yandex.ru

**Artem Babenko**
Yandex
National Research University
Higher School of Economics
artem.babenko@phystech.edu

## ABSTRACT

Nowadays, deep neural networks (DNNs) have become the main instrument for machine learning tasks within a wide range of domains, including vision, NLP, and speech. Meanwhile, in an important case of heterogenous tabular data, the advantage of DNNs over shallow counterparts remains questionable. In particular, there is no sufficient evidence that deep learning machinery allows constructing methods that outperform gradient boosting decision trees (GBDT), which are often the top choice for tabular problems. In this paper, we introduce *Neural Oblivious Decision Ensembles (NODE)*, a new deep learning architecture, designed to work with any tabular data. In a nutshell, the proposed NODE architecture generalizes ensembles of oblivious decision trees, but benefits from both end-to-end gradient-based optimization and the power of multi-layer hierarchical representation learning. With an extensive experimental comparison to the leading GBDT packages on a large number of tabular datasets, we demonstrate the advantage of the proposed NODE architecture, which outperforms the competitors on most of the tasks. We open-source the PyTorch implementation of NODE and believe that it will become a universal framework for machine learning on tabular data.

## 1 INTRODUCTION

The recent rise of deep neural networks (DNN) resulted in a substantial breakthrough for a large number of machine learning tasks in computer vision, natural language processing, speech recognition, reinforcement learning (Goodfellow et al., 2016). Both gradient-based optimization via back-propagation (Rumelhart et al., 1985) and hierarchical representation learning appear to be crucial in increasing the performance of machine learning for these problems by a large margin.

While the superiority of deep architectures in these domains is undoubtful, machine learning for tabular data still did not fully benefit from the DNN power. Namely, the state-of-the-art performance in problems with tabular heterogeneous data is often achieved by "shallow" models, such as gradient boosted decision trees (GBDT) (Friedman, 2001; Chen & Guestrin, 2016; Ke et al., 2017; Prokhorenkova et al., 2018). While the importance of deep learning on tabular data is recognized by the ML community, and many works address this problem (Zhou & Feng, 2017; Yang et al., 2018; Miller et al., 2017; Lay et al., 2018; Feng et al., 2018; Ke et al., 2018), the proposed DNN approaches do not consistently outperform the state-of-the-art shallow models by a notable margin. In particular, to the best of our knowledge, there is still no universal DNN approach that was shown to systematically outperform the leading GBDT packages (e.g., XGBoost (Chen & Guestrin, 2016)). As additional evidence, a large number of Kaggle ML competitions with tabular data are still won by the shallow GBDT methods (Harasymiv, 2015). Overall, at the moment, there is no dominant deep learning solution for tabular data problems, and we aim to reduce this gap by our paper.

We introduce *Neural Oblivious Decision Ensembles (NODE)*, a new DNN architecture, designed to work with tabular problems. The NODE architecture is partially inspired by the recent CatBoost package (Prokhorenkova et al., 2018), which was shown to provide state-of-the-art performance on a large number of tabular datasets. In a nutshell, CatBoost performs gradient boosting on oblivious decision trees (decision tables) (Kohavi, 1994; Lou & Obukhov, 2017), which makes inference very efficient, and the method is quite resistant to overfitting. In its essence, the proposed NODE architecture generalizes CatBoost, making the splitting feature choice and decision tree routing differentiable. As a result, the NODE architecture is fully differentiable and could be incorporated in any computational graph of existing DL packages, such as TensorFlow or PyTorch. Furthermore, NODE allows constructing multi-layer architectures, which resembles "deep" GBDT that is trained end-to-end, which was never proposed before. Besides the usage of oblivious decision tables, another important design choice is the recent *entmax* transformation (Peters et al., 2019), which effectively performs a "soft" splitting feature choice in decision trees inside the NODE architecture. As discussed in the following sections, these design choices are critical to obtain state-of-the-art performance. In a large number of experiments, we compare the proposed approach with the leading GBDT implementations with tuned hyperparameters and demonstrate that NODE outperforms competitors consistently on most of the datasets.

Overall, the main contributions of our paper can be summarized as follows:

1. We introduce a new DNN architecture for machine learning on tabular data. To the best of our knowledge, our method is the first successful example of deep architectures that substantially outperforms leading GBDT packages on tabular data.
2. Via an extensive experimental evaluation on a large number of datasets, we show that the proposed NODE architecture outperforms existing GBDT implementations.
3. The PyTorch implementation of NODE is available online[1].

The rest of the paper is organized as follows. In Section 2 we review prior work relevant to our method. The proposed Neural Oblivious Decision Ensembles architecture is described in Section 3 and experimentally evaluated in Section 4. Section 5 concludes the paper.

## 2 RELATED WORK

In this section, we briefly review the main ideas from prior work that are relevant to our method.

**The state-of-the-art for tabular data.** Ensembles of decision trees, such as GBDT (Friedman, 2001) or random forests (Barandiaran, 1998), are currently the top choice for tabular data problems. Currently, there are several leading GBDT packages, such as XGBoost (Chen & Guestrin, 2016), LightGBM (Ke et al., 2017), CatBoost (Prokhorenkova et al., 2018), which are widely used by both academicians and ML practitioners. While these implementations vary in details, on most of the tasks their performances do not differ much (Prokhorenkova et al., 2018; Anghel et al.). The most important distinction of CatBoost is that it uses oblivious decision trees (ODTs) as weak learners. As ODTs are also an important ingredient of our NODE architecture, we discuss them below.

**Oblivious Decision Trees.** An oblivious decision tree is a regular tree of depth $d$ that is constrained to use the same splitting feature and splitting threshold in all internal nodes of the same depth. This constraint essentially allows representing an ODT as a table with $2^d$ entries, corresponding to all possible combinations of $d$ splits (Lou & Obukhov, 2017). Of course, due to the constraints above, ODTs are significantly weaker learners compared to unconstrained decision trees. However, when used in an ensemble, such trees are less prone to overfitting, which was shown to synergize well with gradient boosting (Prokhorenkova et al., 2018). Furthermore, the inference in ODTs is very efficient: one can compute $d$ independent binary splits in parallel and return the appropriate table entry. In contrast, non-oblivious decision trees require evaluating $d$ splits sequentially.

**Differentiable trees.** The significant drawback of tree-based approaches is that they usually do not allow end-to-end optimization and employ greedy, local optimization procedures for tree construction. Thus, they cannot be used as a component for pipelines, trained in an end-to-end fashion. To address this issue, several works (Kontschieder et al., 2015; Yang et al., 2018; Lay et al., 2018)

---

[1]https://github.com/Qwicen/node

propose to "soften" decision functions in the internal tree nodes to make the overall tree function and tree routing differentiable. In our work, we advocate the usage of the recent entmax transformation (Peters et al., 2019) to "soften" decision trees. We confirm its advantages over the previously proposed approaches in the experimental section.

**Entmax.** The key building block of our model is the entmax transformation (Peters et al., 2019), which maps a vector of real-valued scores to a discrete probability distribution. This transformation generalizes the traditional softmax and its sparsity-enforcing alternative sparsemax (Martins & Astudillo, 2016), which has already received significant attention in a wide range of applications: probabilistic inference, topic modeling, neural attention (Niculae & Blondel, 2017; Niculae et al., 2018; Lin et al., 2019). The entmax is capable to produce sparse probability distributions, where the majority of probabilities are exactly equal to $0$. In this work, we argue that entmax is also an appropriate inductive bias in our model, which allows differentiable split decision construction in the internal tree nodes. Intuitively, entmax can learn splitting decisions based on a small subset of data features (up to one, as in classical decision trees), avoiding undesired influence from others. As an additional advantage, using entmax for feature selection allows for computationally efficient inference using the sparse pre-computed choice vectors as described below in Section 3.

**Multi-layer non-differentiable architectures.** Another line of work (Miller et al., 2017; Zhou & Feng, 2017; Feng et al., 2018) promotes the construction of multi-layer architectures from non-differentiable blocks, such as random forests or GBDT ensembles. For instance, (Zhou & Feng, 2017; Miller et al., 2017) propose to use stacking of several random forests, which are trained separately. In recent work, (Feng et al., 2018) introduces the multi-layer GBDTs and proposes a training procedure that does not require each layer component to be differentiable. While these works report marginal improvements over shallow counterparts, they lack the capability for end-to-end training, which could result in inferior performance. In contrast, we argue that end-to-end training is crucial and confirm this claim in the experimental section.

**Specific DNN for tabular data.** While a number of prior works propose architectures designed for tabular data (Ke et al., 2018; Shavitt & Segal, 2018), they mostly do not compare with the properly tuned GBDT implementations, which are the most appropriate baselines. The recent preprint (Ke et al., 2018) reports the marginal improvement over GBDT with default parameters, but in our experiments, the baseline performance is much higher. To the best of our knowledge, our approach is the first to consistently outperform the tuned GBDTs over a large number of datasets.

## 3 NEURAL OBLIVIOUS DECISION ENSEMBLES

We introduce the Neural Oblivious Decision Ensemble (NODE) architecture with a layer-wise structure similar to existing deep learning models. In a nutshell, our architecture consists of differentiable oblivious decision trees (ODT) that are trained end-to-end by backpropagation. We describe our implementation of the differentiable NODE layer in Section 3.1, the full model architecture in Section 3.2, and the training and inference procedures in section 3.3.

### 3.1 DIFFERENTIABLE OBLIVIOUS DECISION TREES

The core building block of our model is a Neural Oblivious Decision Ensemble (NODE) layer. The layer is composed of $m$ differentiable oblivious decision trees (ODTs) of equal depth $d$. As an input, all $m$ trees get a common vector $x \in \mathbb{R}^n$, containing $n$ numeric features. Below we describe a design of a single differentiable ODT.

In its essence, an ODT is a decision table that splits the data along $d$ splitting features and compares each feature to a learned threshold. Then, the tree returns one of the $2^d$ possible responses, corresponding to the comparisons result. Therefore, each ODT is completely determined by its splitting features $f \in \mathbb{R}^d$, splitting thresholds $b \in \mathbb{R}^d$ and a $d$-dimensional tensor of responses $R \in \mathbb{R}^{\overbrace{2 \times 2 \times 2}^{d}}$. In this notation, the tree output is defined as:

$$h(x) = R[\mathbb{1}(f_1(x) - b_1), \ldots, \mathbb{1}(f_d(x) - b_d)], \qquad (1)$$

where $\mathbb{1}(\cdot)$ denotes the Heaviside function.

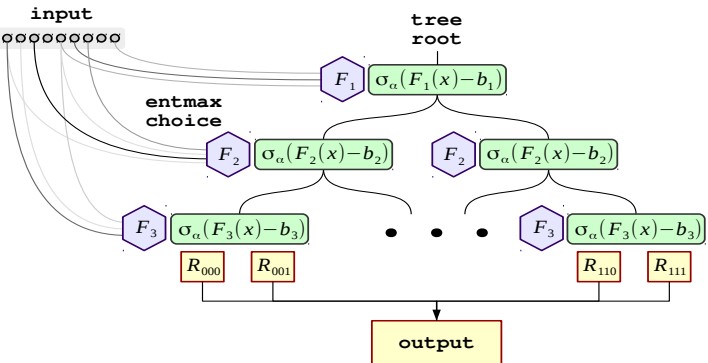

Figure 1: The single ODT inside the NODE layer. The splitting features and the splitting thresholds are shared across all the internal nodes of the same depth. The output is a sum of leaf responses scaled by the choice weights.

To make the tree output (1) differentiable, we replace the splitting feature choice $f_i$ and the comparison operator $\mathbb{1}(f_i(x) - b_i)$ by their continuous counterparts. There are several existing approaches that can be used for modelling differentiable choice functions in decision trees (Yang et al., 2018), for instance, REINFORCE (Williams, 1992) or Gumbel-softmax (Jang et al., 2016). However, these approaches typically require long training time, which can be crucial in practice.

Instead, we propose to use the $\alpha$-entmax function (Peters et al., 2019) as it is able to learn sparse choices, depending only on a few features, via standard gradient descent. This function is a generalization of softmax in its variational form: $softmax(x) = argmax_{p \in \Delta}[\langle p, x \rangle + H(p)]$ where $H(p)$ is Shannon entropy. We can define $\alpha$-entmax by replacing $H(p)$ with Tsallis $\alpha$-entropy[2].

The choice function is hence replaced by a weighted sum of features, with weights computed as $\alpha$-entmax ($\alpha$=1.5) over the learnable feature selection matrix $F \in \mathbb{R}^{d \times n}$:

$$\hat{f}_i(x) = \sum_{j=1}^{n} x_j \cdot entmax_\alpha(F_{ij}) \tag{2}$$

Similarly, we relax the Heaviside function $\mathbb{1}(f_i(x) - b_i)$ as a two-class entmax, which we denote as $\sigma_\alpha(x) = entmax_\alpha([x, 0])$. As different features can have different characteristic scales, we use the scaled version $c_i(x) = \sigma_\alpha\left(\frac{f_i(x) - b_i}{\tau_i}\right)$, where $b_i$ and $\tau_i$ are learnable parameters for thresholds and scales respectively.

Based on the $c_i(x)$ values, we define a "choice" tensor $C \in \mathbb{R}^{\overbrace{2 \times 2 \times 2}^{d}}$ of the same size as the response tensor $R$ by computing the outer product of all $c_i$:

$$C(x) = \begin{bmatrix} c_1(x) \\ 1 - c_1(x) \end{bmatrix} \otimes \begin{bmatrix} c_2(x) \\ 1 - c_2(x) \end{bmatrix} \otimes \cdots \otimes \begin{bmatrix} c_d(x) \\ 1 - c_d(x) \end{bmatrix} \tag{3}$$

The final prediction is then computed as a weighted linear combination of response tensor entries $R$ with weights from the entries of choice tensor $C$:

$$\hat{h}(x) = \sum_{i_1, \ldots i_d \in \{0,1\}^d} R_{i_1, \ldots, i_d} \cdot C_{i_1, \ldots, i_d}(x) \tag{4}$$

Note, that this relaxation equals to the classic non-differentiable ODT $h(x)(1)$ iff both feature selection and threshold functions reach one-hot state, i.e. entmax always returns non-zero weights for a single feature and $c_i$ always return exactly zeros or ones.

---

[2]If one is unfamiliar with this definition, we highly recommend reading Peters et al. (2019)

Finally, the output of the NODE layer is composed as a concatenation of the outputs of $m$ individual trees $\left[\hat{h}_1(x), \ldots, \hat{h}_m(x)\right]$.

**Multidimensional tree outputs.** In the description above, we assumed that tree outputs are one-dimensional $\hat{h}(x) \in \mathbb{R}$. For classification problems, where NODE predicts probabilities of each class, we use multidimensional tree outputs $\hat{h}(x) \in \mathbb{R}^{|C|}$, where $|C|$ is a number of classes.

## 3.2 GOING DEEPER WITH THE NODE ARCHITECTURE

The NODE layer, described above, can be trained alone or within a complex structure, like fully-connected layers that can be organized into the multi-layer architectures. In this work, we introduce a new architecture, following the popular DenseNet (Huang et al., 2017) model and train it end-to-end via backpropagation.

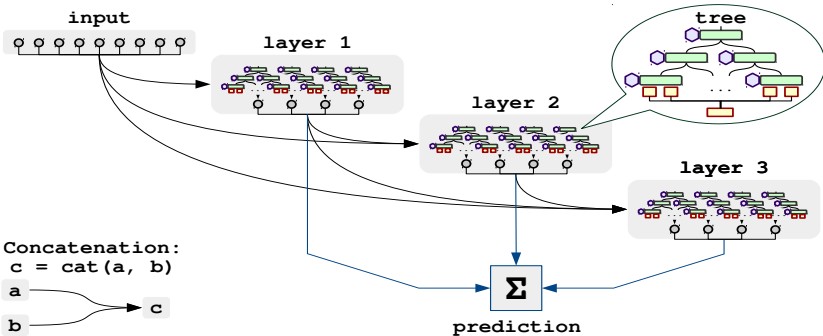

Figure 2: The NODE architecture, consisting of densely connected NODE layers. Each layer contains several trees whose outputs are concatenated and serve as input for the subsequent layer. The final prediction is obtained by averaging the outputs of all trees from all the layers.

Similar to DenseNet, our architecture is a sequence of $k$ NODE layers (see Section 3.1), where each layer uses a concatenation of all previous layers as its input. The input layer 0 of this architecture corresponds to the input features $x$, accessible by all successor layers. Due to such a design, our architecture is capable to learn both shallow and deep decision rules. A single tree on $i$-th layer can rely on chains of up to $i-1$ layer outputs as features, allowing it to capture complex dependencies. The resulting prediction is a simple average of all decision trees from all layers.

Note, in the multi-layer architecture described above, tree outputs $\hat{h}(x)$ from early layers are used as inputs for subsequent layers. Therefore, we do not restrict the dimensionality of $\hat{h}(x)$ to be equal to the number of classes, and allow it to have an arbitrary dimensionality $l$, which correspond to the $(d+1)$-dimensional response tensor $R \in \mathbb{R}^{\underbrace{2 \times 2 \times 2}_{d} \times l}$. When averaging the predictions from all layers, only first $|C|$ coordinates of $\hat{h}(x)$ are used for classification problems and the first one for regression problems. Overall, $l$ is an additional hyperparameter with typical values in $[1, 3]$.

## 3.3 TRAINING

Here we summarize the details of our training protocol.

**Data preprocessing.** First, we transform each data feature to follow a normal distribution via quantile transform[3]. In experiments, we observed that this step was important for stable training and faster convergence.

**Initialization.** Before training, we perform the data-aware initialization (Mishkin & Matas, 2016) to obtain a good initial parameter values. In particular, we initialize the feature selection matrix

---

[3]sklearn.preprocessing.QuantileTransformer

uniformly $F_{ij} \sim U(0, 1)$, while the thresholds $b$ are initialized with random feature values $f_i(x)$ observed in the first data batch. The scales $\tau_i$ are initialized in such a way that all the samples in the first batch belong to the linear region of $\sigma_\alpha$, and hence receive nonzero gradients. Finally, the response tensor entries are initialized with the standard normal distribution $R[i_1, \ldots, i_d] \sim N(0, 1)$.

**Training.** As for existing DNN architectures, NODE is trained end-to-end via mini-batch SGD. We jointly optimize all model parameters: $F, b, R$. In this work, we experimented with traditional objective functions (cross-entropy for classification and mean squared error for regression), but any differentiable objective can be used as well. As an optimization method, we use the recent Quasi-Hyperbolic Adam with parameters recommended in the original paper (Ma & Yarats, 2018). We also average the model parameters over $c = 5$ consecutive checkpoints (Izmailov et al., 2018) and pick the optimal stopping point on the hold-out validation dataset.

**Inference.** During training, a significant fraction of time is spent computing the entmax function and multiplying the choice tensor. Once the model is trained, one can pre-compute entmax feature selectors and store them as a sparse vector (e.g., in coordinate (coo) format), making inference more efficient.

## 4 EXPERIMENTS

In this section, we report the results of a comparison between our approach and the leading GBDT packages. We also provide several ablation studies that demonstrate the influence of each design choice in the proposed NODE architecture.

### 4.1 COMPARISON TO THE STATE-OF-THE-ART.

As our main experiments, we compare the proposed NODE architecture with two state-of-the-art GBDT implementations on a large number of datasets. In all the experiments we set $\alpha$ parameter in the entmax transformation to $1.5$. All other details of the comparison protocol are described below.

**Datasets.** We perform most of the experiments on six open-source tabular datasets from different domains: **Epsilon, YearPrediction, Higgs, Microsoft, Yahoo, Click**. The detailed description of the datasets is available in appendix. All the datasets provide train/test splits, and we used 20% samples from the train set as a validation set to tune the hyperparameters. For each dataset, we fix the train/val/test splits for a fair comparison. For the classification datasets (Epsilon, Higgs, Click), we minimize cross-entropy loss and report the classification error. For the regression and ranking datasets (YearPrediction, Microsoft, Yahoo), we minimize and report mean squared error (which corresponds to the pointwise approach to learning-to-rank).

**Methods.** We compare the proposed NODE architecture to the following baselines:

- **Catboost.** The recent GBDT implementation (Prokhorenkova et al., 2018) that uses oblivious decision trees as weak learners. We use the open-source implementation, provided by the authors.
- **XGBoost.** The most popular GBDT implementation widely used in machine learning competitions (Chen & Guestrin, 2016). We use the open-source implementation, provided by the authors.
- **FCNN.** Deep neural network, consisting of several fully-connected layers with ReLU non-linearity layers (Nair & Hinton, 2010).

**Regimes.** We perform comparison in two following regimes that are the most important in practice:

- **Default hyperparameters.** In this regime, we compare the methods as easy-to-tune toolkits that could be used by a non-professional audience. Namely, here we do not tune hyperparameters and use the default ones provided by the GBDT packages. The only tunable parameter here is a number of trees (up to 2048) in CatBoost/XGBoost, which is set based on the validation set. We do not compare with FCNN in this regime, as it typically requires much tuning, and we did not find the set of parameters, appropriate for all datasets. The default architecture in our model contains only a single layer with 2048 decision trees of depth six. Both of these hyperparameters were inherited from the CatBoost package settings for oblivious decision trees. With these parameters, the NODE architecture is shallow, but it still benefits from end-to-end training via back-propagation.

|  | Epsilon | YearPrediction | Higgs | Microsoft | Yahoo | Click |
|---|---|---|---|---|---|---|
| **Default hyperparameters** | | | | | | |
| CatBoost | $0.1119\pm2e-4$ | $80.68\pm0.04$ | $0.2434\pm2e-4$ | $0.5587\pm2e-4$ | $0.5781\pm3e-4$ | $0.3438\pm1e-3$ |
| XGBoost | 0.1144 | 81.11 | 0.2600 | 0.5637 | 0.5756 | 0.3461 |
| NODE | **$0.1043\pm4e-4$** | **$77.43\pm0.09$** | **$0.2412\pm5e-4$** | **$0.5584\pm3e-4$** | **$0.5666\pm5e-4$** | **$0.3309\pm3e-4$** |

Table 1: The comparison of NODE with the shallow state-of-the-art counterparts with default hyperparameters. The results are computed over ten runs with different random seeds.

|  | Epsilon | YearPrediction | Higgs | Microsoft | Yahoo | Click |
|---|---|---|---|---|---|---|
| **Tuned hyperparameters** | | | | | | |
| CatBoost | $0.1113\pm4e-4$ | $79.67\pm0.12$ | $0.2378\pm1e-4$ | $0.5565\pm2e-4$ | $0.5632\pm3e-4$ | $0.3401\pm2e-3$ |
| XGBoost | $0.1112\pm6e-4$ | $78.53\pm0.09$ | $0.2328\pm3e-4$ | **$0.5544\pm1e-4$** | **$0.5420\pm4e-4$** | $0.3334\pm2e-3$ |
| FCNN | $0.1058\pm1e-3$ | $81.17\pm0.81$ | $0.2139\pm4e-4$ | $0.5595\pm2e-4$ | $0.5691\pm7e-4$ | $0.3324\pm6e-5$ |
| NODE | **$0.1034\pm3e-4$** | **$76.21\pm0.12$** | **$0.2101\pm5e-4$** | $0.5570\pm2e-4$ | $0.5692\pm2e-4$ | **$0.3312\pm2e-3$** |
| mGBDT | OOM | 80.67 | OOM | OOM | OOM | OOM |
| DeepForest | 0.1179 | — | 0.2391 | — | — | 0.3333 |

Table 2: The comparison of NODE with both shallow and deep counterparts with hyperparameters tuned for optimal performance. The results are computed over ten runs with different random seeds.

- **Tuned hyperparameters.** In this regime, we tune the hyperparameters for both NODE and the competitors on the validation subsets. The optimal configuration for NODE contains between two and eight NODE layers, while the total number of trees across all the layers does not exceed 2048. The details of hyperparameter optimization are provided in appendix.

The results of the comparison are summarized in Table 1 and Table 2. For all methods, we report mean performance and standard deviations computed over ten runs with different random seeds. Several key observations are highlighted below:

1. With default hyperparameters, the proposed NODE architecture consistently outperforms both CatBoost and XGBoost on all datasets. The results advocate the usage of NODE as a handy tool for machine learning on tabular problems.
2. With tuned hyperparameters, NODE also outperforms the competitors on most of the tasks. Two exceptions are the **Yahoo** and **Microsoft** datasets, where tuned XGBoost provides the highest performance. Given the large advantage of XGBoost over CatBoost on **Yahoo**, we speculate that the usage of oblivious decision trees is an inappropriate inductive bias for this dataset. This implies that NODE should be extended to non-oblivious trees, which we leave for future work.
3. In the regime with tuned hyperparameters on some datasets FCNN outperforms GBDT, while on others GBDT is superior. Meanwhile, the proposed NODE architecture appears to be a universal instrument, providing the highest performance on most of the tasks.

For completeness we also aimed to compare to previously proposed architectures for deep learning on tabular data. Unfortunately, many works did not publish the source code. We were only able to perform a partial comparison with mGBDT (Feng et al., 2018) and DeepForest (Zhou & Feng, 2017), which source code is available. For both baselines, we use the implementations, provided by the authors, and tune the parameters on the validation set. Note, that the DeepForest implementation is available only for classification problems. Moreover, both implementations do not scale well, and for many datasets, we obtained Out-Of-Memory error (OOM). On datasets in our experiments it turns out that properly tuned GBDTs outperform both (Feng et al., 2018) and (Zhou & Feng, 2017).

## 4.2 ABLATIVE ANALYSIS

In this section, we analyze the key architecture components that define our model.

**Choice functions.** Constructing differentiable decision trees requires a function that selects items from a set. Such function is required for both splitting feature selection and decision tree routing. We experimented with four possible options, each having different implications:

| Dataset | YearPrediction | | | | Epsilon | | | |
|---|---|---|---|---|---|---|---|---|
| Function | softmax | Gumbel | sparsemax | entmax | softmax | Gumbel | sparsemax | entmax |
| 1 layer | 78.41 | 79.39 | 78.13 | 77.43 | 0.1045 | 0.1979 | 0.1083 | 0.1043 |
| 2 layers | 77.61 | 79.31 | 76.81 | 77.05 | 0.1041 | 0.2884 | 0.1052 | 0.1031 |
| 4 layers | 77.58 | 79.69 | 76.60 | 76.21 | 0.1034 | 0.2908 | 0.1058 | 0.1033 |
| 8 layers | 77.47 | 80.49 | 76.31 | 76.17 | 0.1036 | 0.3081 | 0.1058 | 0.1036 |

Table 3: The experimental comparison of various choice functions and architecture depths. The values represent mean squared error for YearPrediction and classification error rate for Epsilon.

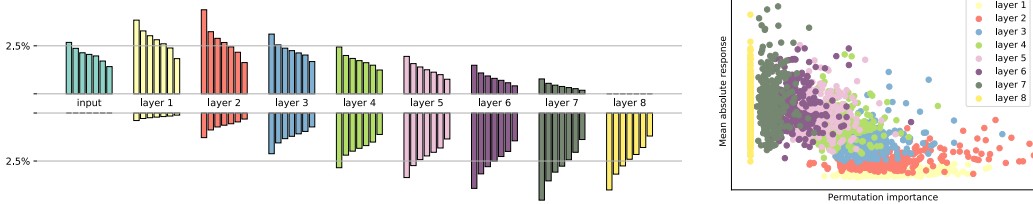

Figure 3: NODE on UCI Higgs dataset: **Left-Top:** individual feature importance distributions for both original and learned features. **Left-Bottom:** mean absolute contribution of individual trees to the final response. **Right:** responses dependence on feature importances. See details in the text.

- **Softmax** learns dense decision rules where all items have nonzero weights;
- **Gumbel-Softmax (Jang et al., 2016)** learns to stochastically sample a single element from a set;
- **Sparsemax (Martins & Astudillo, 2016)** learns sparse decision rules, where only a few items have nonzero weights;
- **Entmax (Peters et al., 2019)** generalizes both sparsemax and softmax; it is able to learn sparse decision rules, but is smoother than sparsemax, being more appropriate for gradient-based optimization. In comparison $\alpha$ parameter was set to $1.5$.

We experimentally compare the four options above with both shallow and deep architectures in Table 3. We use the same choice function for both feature selection and tree routing across all experiments. In Gumbel-Softmax, we replaced it with hard argmax one-hot during inference. The results clearly show that Entmax with $\alpha=1.5$ outperforms the competitors across all experiments. First, Table 3 demonstrates that sparsemax and softmax are not universal choice functions. For instance, on the YearPrediction dataset, sparsemax outperforms softmax, while on the Epsilon dataset softmax is superior. In turn, entmax provides great empirical performance across all datasets. Another observation is that Gumbel-Softmax is unable to learn deep architectures with both constant and annealed temperature schedules. This behavior is probably caused by the stochasticity of Gumbel-Softmax and the responses on the former layers are too noisy to produce useful features for the latter layers.

**Feature importance.** In this series of experiments, we analyze the internal representations, learned by the NODE architecture. We begin by estimating the feature importances from different layers of a multi-layer ensemble via permutation feature importance, initially introduced in (Breiman, 2001). Namely, for 10.000 objects from the Higgs dataset we randomly shuffle the values of each feature (original or learnt on some NODE layer) and compute the increase in the classification error. Then for each layer, we split feature importance values into seven equal bins and calculate the total feature importance of each bin, shown on Figure 3 (left-top). We discovered that the features from the first layer are used the most, with feature importances decreasing with depth. This figure shows that deep layers are able to produce important features, even though earlier layers have an advantage because of the DenseNet architecture. Next, we estimated the mean absolute contribution of individual trees to the final response, reported on Figure 3 (left-bottom). One can see the reverse trend, deep trees tend to contribute more to the final response. Figure 3 (right) clearly shows that there is anticorrelation of feature importances and contributions in the final response, which implies that the main role of ealier layers is to produce informative features, while the latter layers mostly use them for accurate prediction.

**Training/Inference runtime.** Finally, we compare the NODE runtime to the timings of the state-of-the-art GBDT implementations. In Table 4 we report the training and inference time for million of objects from the YearPrediction dataset. In this experiment, we evaluate ensembles of 1024 trees

of depth six with all other parameters set to their default values. Our GPU setup has a single 1080Ti GPU and 2 CPU cores. In turn, our CPU setup has a 28-core Xeon E5-2660 v4 processor (which costs almost twice as much as the GPU). We use CatBoost v0.15 and XGBoost v0.90 as baselines, while NODE inference runs on PyTorch v1.1.0. Overall, NODE inference time is on par with heavily optimized GBDT libraries despite being implemented in pure PyTorch (i.e. no custom kernels).

| Method | NODE 8 layers 1080Ti | XGBoost Xeon | XGBoost 1080Ti | CatBoost Xeon |
|---|---|---|---|---|
| Training | 7min 42s | 5min 39s | 1min 13s | 41s |
| Inference | 8.56s | 5.94s | 4.45s | 4.62s |

Table 4: Training and inference runtime for models with 1024 trees of depth six on the YearPrediction dataset, averaged over five runs. Both training and inference of eight-layer NODE architecture on GPU are on par with shallow counterparts of the same total number of trees in an ensemble.

## 5  CONCLUSION

In this paper, we introduce a new DNN architecture for deep learning on heterogeneous tabular data. The architecture is differentiable deep GBDTs, trained end-to-end via backpropagation. In extensive experiments, we demonstrate the advantages of our architecture over existing competitors with the default and tuned hyperparameters. A promising research direction is incorporating the NODE layer into complex pipelines trained via back-propagation. For instance, in multi-modal problems, the NODE layer could be employed as a way to incorporate the tabular data, as CNNs are currently used for images, or RNNs are used for sequences.

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

| | Train | Test | Features | Task | Metric | Description |
|---|---|---|---|---|---|---|
| Epsilon[5] | 400K | 100K | 2000 | Classification | Error | PASCAL Challenge 2008 |
| YearPrediction[6] | 463K | 51.6K | 90 | Regression | MSE | Million Song Dataset |
| Higgs[7] | 10.5M | 500K | 28 | Classification | Error | UCI ML Higgs |
| Microsoft[8] | 723K | 241K | 136 | Regression | MSE | MSLR-WEB10K |
| Yahoo[9] | 544K | 165K | 699 | Regression | MSE | Yahoo LETOR dataset |
| Click[10] | 800K | 200K | 11 | Classification | Error | 2012 KDD Cup |

Table 5: The datasets used in our experiments.

## A    APPENDIX

### A.1    DESCRIPTION OF THE DATASETS

In our experiments, we used six tabular datasets, described in Table 5. (1) Epsilon is high dimensional dataset from the PASCAL Large Scale Learning Challenge 2008. The problem is a binary classification. (2) YearPrediction is a subset of Million Song Dataset. It is regression dataset, and the task is to predict the release year of the song by using the audio features. It contains tracks from 1922 to 2011. (3) Higgs is a dataset from the UCI ML Repository. The problem is to predict whether the given event produces Higgs bosons or not. (4) Microsoft is a Learning to Rank Dataset. It consists of 136-dimensional feature vectors extracted from query-url pairs. Each pair has relevance judgment labels, which take values from 0 (irrelevant) to 4 (perfectly relevant) (5) Yahoo is very similar ranking dataset with query-url pairs labeled from 0 to 4. We treat both ranking problems as regression (which corresponds to the pointwise approach to learning-to-rank) (6) Click is a subset of data from the 2012 KDD Cup. For the subset construction, we randomly sample 500.000 objects of a positive class and 500.000 objects of a negative class. The categorical features were converted to numerical ones via Leave-One-Out encoder from category_encoders package of the scikit-learn library.

### A.2    OPTIMIZATION OF HYPERPARAMETERS

In order to tune the hyperparameters, we performed a random stratified split of full training data into train set (80%) and validation set (20%) for the Epsilon, YearPrediction, Higgs, Microsoft, and Click datasets. For Yahoo, we use train/val/test split provided by the dataset authors. We use the Hyperopt[4] library to optimize Catboost, XGBoost, and FCNN hyperparameters. For each method, we perform 50 steps of Tree-structured Parzen Estimator (TPE) optimization algorithm. As a final configuration, we choose the set of hyperparameters, corresponding to the smallest loss on the validation set.

### A.2.1    CATBOOST AND XGBOOST

On each iteration of Hyperopt, the number of trees was set based on the validation set, with maximal trees count set to 2048. Below is the list of hyperparameters and their search spaces for Catboost.

- *learning_rate*: Log-Uniform distribution $[e^{-5}, 1]$

- *random_strength*: Discrete uniform distribution $[1, 20]$

- *one_hot_max_size*: Discrete uniform distribution $[0, 25]$

- *l2_leaf_reg*: Log-Uniform distribution $[1, 10]$

- *bagging_temperature*: Uniform distribution $[0, 1]$

- *leaf_estimation_iterations*: Discrete uniform distribution $[1, 10]$

---

[4]https://github.com/hyperopt/hyperopt

XGBoost tuned parameters and their search spaces:

- *eta*: Log-Uniform distribution $[e^{-7}, 1]$
- *max_depth*: Discrete uniform distribution $[2, 10]$
- *subsample*: Uniform distribution $[0.5, 1]$
- *colsample_bytree*: Uniform distribution $[0.5, 1]$
- *colsample_bylevel*: Uniform distribution $[0.5, 1]$
- *min_child_weight*: Log-Uniform distribution $[e^{-16}, e^5]$
- *alpha*: Uniform choice $\{0$, Log-Uniform distribution $[e^{-16}, e^2]\}$
- *lambda*: Uniform choice $\{0$, Log-Uniform distribution $[e^{-16}, e^2]\}$
- *gamma*: Uniform choice $\{0$, Log-Uniform distribution $[e^{-16}, e^2]\}$

### A.2.2 FCNN

Fully connected neural networks were tuned using Hyperas [11] library, which is a Keras wrapper for Hyperopt. We consider FCNN constructed from the following blocks: Dense-ReLU-Dropout. The number of units in each layer is independent of each other, and dropout value is the same for the whole network. The networks are trained with the Adam optimizer with averaging the model parameters over $c=5$ consecutive checkpoints (Izmailov et al., 2018) and early stopping on validation. Batch size is fixed to 1024 for all datasets. Below is the list of tuned hyperparameters.

- *Architecture:* either Sequential (each layer is receives previous layer activations) or DenseNet (each layer receives activations from all previous layers
- *Number of layers*: Discrete uniform distribution $[2, 7]$
- *Number of units*: Dicrete uniform distribution over a set $\{128, 256, 512, 1024\}$
- *Learning rate*: Uniform distribution $[1e - 4, 1e - 2]$
- *Dropout*: Uniform distribution $[0, 0.5]$

### A.2.3 NODE

Neural Oblivious Decision Ensembles were tuned by grid search over the following hyperparameter values. In the multi-layer NODE, we use the same architecture for all layers, i.e., the same number of trees of the same depth. *total_tree_count* here denotes the total number of trees on all layers. For each dataset, we use the maximal batch size, which fits in the GPU memory. We always use learning rate $10^{-3}$.

- *num_layers*: $\{2, 4, 8\}$
- *total_tree_count*: $\{1024, 2048\}$
- *tree_depth*: $\{6, 8\}$
- *tree_output_dim*: $\{2, 3\}$

---

[5]https://www.csie.ntu.edu.tw/ cjlin/libsvmtools/datasets/binary.html

[6]https://archive.ics.uci.edu/ml/datasets/yearpredictionmsd

[7]https://archive.ics.uci.edu/ml/datasets/HIGGS

[8]https://www.microsoft.com/en-us/research/project/mslr/

[9]https://webscope.sandbox.yahoo.com/catalog.php?datatype=c

[10]http://www.kdd.org/kdd-cup/view/kdd-cup-2012-track-2

[11]https://github.com/maxpumperla/hyperas

