# OpenReview forum: "Neural Oblivious Decision Ensembles for Deep Learning on Tabular Data"
_ICLR.cc/2020/Conference — Accept (Poster)_

### Official Review · AnonReviewer1 · 2019-10-22
**Official Blind Review #1**

**Rating:** 6

**Review:**

The paper tries to ask if there is a good neural net architecture that works as effectively as gradient boosting decision trees on tabular data. The authors propose an architecture (NODE) that satisfies this conditions. NODE is an architecture consisting of differentiable oblivious decision trees that can be trained end to end via back propagation. The paper is readable and the experiments are well presented. They make use of an alpha-entmax transformation to obtain a differentiable architecture. The approach seems well motivated in the literature. It is unclear how novel the contribution is. It is unclear if in the experimental section the datasets used are standard for this classes of tasks. Would be good to mention if it is the case.

**Experience Assessment:**

I do not know much about this area.

**Review Assessment: Checking Correctness Of Derivations And Theory:**

I did not assess the derivations or theory.

**Review Assessment: Checking Correctness Of Experiments:**

I assessed the sensibility of the experiments.

**Review Assessment: Thoroughness In Paper Reading:**

I made a quick assessment of this paper.

---

> ### Author Response · Authors · 2019-11-09
> **R#1: We try to address your concerns.**
>
> We thank you for the review.
>
> [It is unclear if in the experimental section the datasets used are standard for this classes of tasks.]
> All datasets from our experiments are standard for tabular data processing: each dataset was previously featured in multiple published studies. We deliberately chose these six datasets to cover different domain areas [web, natural sciences, etc.], tasks [classification/regression] and dataset sizes.

---

### Official Review · AnonReviewer2 · 2019-10-24
**Official Blind Review #2**

**Rating:** 8

**Review:**

Paper Summary:

The paper considers training oblivious trees ensemble with gradient descent by introducing a relaxation for feature selection and node thresholding. The relaxation is based on the recently introduced EntMax. The approach is compared with standard gradient boosting tree learning on benchmark datasets.

Review Summary:

The paper reads well, is technically sound. The approach is novel and relevant to ICLR. Reference to related work are appropriate. Experimental comparison with CatBoost, neural nets could be more rigorous, more ablations could give a complete picture. Overall this is a good paper that gives an extra tool applicable to many practical settings.

Detailed Review:

The introduction needs to define "tabular data". In your case, it seems that you mean mostly numerical heterogeneous features. Could you comment on using categorical features as well?

The method is clearly explained and references are appropriate, so most of my questions relate to the empirical setup and results.

First, it seems to me that the paper would be much stronger if you were to reproduce the results from an established paper. If you take the catboost paper (arXiv:1706.09516v5 [cs.LG] 20 Jan 2019), the error on epsilon dataset is 10.9 which is better than the number your report, similarly click reports 15.6 error rate. To me, the paper would be much better if you simply added an FCNN and a NODE column to Table 2 and 3 of the catboost paper. It does not mean that your approach has to be better in all cases, but it will give a clear picture of when it is useful and it would clear any doubt on the tuning of the catboost baseline.

Second, the model you propose builds upon the densenet idea while the FCNN you compare with has no densenet connections. It would be fairer to consider neural net with this kind of residual.

Third, I feel you need to report results over CPU as well. Boosted trees primary advantage is their low cost on regular CPU, the entmax formulation requires integrating over more leaves
than typical thresholded trees and it would be interesting to compare the effect on CPU. Reporting timing with batch and individual sample evaluation would make sense as well.

 As a side note, I would advise to define entmax with its equation. It is too recent to consider it should be known by the reader.

Overall, this is a good paper than reads well. The method is novel, interesting and practical. With the extra experiments, it would make an excellent ICLR paper.

**Experience Assessment:**

I have published one or two papers in this area.

**Review Assessment: Checking Correctness Of Derivations And Theory:**

I carefully checked the derivations and theory.

**Review Assessment: Checking Correctness Of Experiments:**

I carefully checked the experiments.

**Review Assessment: Thoroughness In Paper Reading:**

I read the paper thoroughly.

---

> ### Author Response · Authors · 2019-11-09
> **R#2: We try to address your concerns.**
>
> Thank you for your comments, we address your concerns below.
>
> [add comparison to FCNN with DenseNet connections]
> We agree and conduct an additional set of experiments focused on densely-connected FCNN models. We use the standard FCNN tuning procedure described in the submission. Numbers in the table below correspond to the performance on the val/test subsets.
>
> FCNN              |    Epsilon         | YearPrediction |     Higgs           |   Microsoft      |     Yahoo           |     Click             |
> Sequential     | 0.1041/0.1043 |   70.07/79.99   | 0.2140/0.2140 | 0.5411/0.5608 | 0.5977/0.5773 | 0.3303/0.3325 |
> DenseNet      | 0.1044/0.1043 |   69.00/81.17   | 0.2146/0.2139 | 0.5403/0.5595 | 0.5899/0.5691 | 0.3302/0.3324 |
>
> As you can see, DenseNet does indeed sometimes outperform the sequential architecture but does not outperform NODE, which indicates that the inductive bias of oblivious decision ensembles is important. We have included dense connections in FCNN parameter tuning scheme and have updated Table 2 in a new revision
>
> [it seems to me that the paper would be much stronger if you were to reproduce the results from an established paper.]
>
> We agree with this concern. However, the choice of benchmark datasets in the Catboost paper is biased to categorical features as it is the main focus of Catboost. Moreover, most of the datasets are quite small, see Table 7 in https://arxiv.org/pdf/1706.09516.pdf. In contrast, we aim to cover different dataset sizes and domain areas.
>
> [Third, I feel you need to report results over CPU as well]
> We agree that this would be a valuable addition. However, our pytorch-based implementation specifically targets GPU training and inference. A naive conversion of 8-layer NODE to run on 28-core Xeon E5-2660 v4 has an average training time of 49min 40s and inference time of 1m 4.5s per million predictions on the YearPrediction dataset.
>
> The majority of this time is spent on multiplying activations by zero - a side-effect of a highly parallel GPU-friendly implementation. We expect that in a CPU-optimized NODE implementation (e.g. with natively compiled C++), inference would take between 100% and 200% of CatBoost inference time. However, development of such optimized implementation would take up immense amounts of time and effort and is not possible till the end of discussion period.
>
> [I would advise to define entmax with its equation]
>
> We have described entmax with more details in a new revision.

---

### Official Review · AnonReviewer3 · 2019-10-24
**Official Blind Review #3**

**Rating:** 3

**Review:**

This paper introduces a new method to make ensembles of decision trees differentiable, and trainable with (stochastic) gradient descent. The proposed technique relies on the concept of "oblivious decision trees", which are a kind of decision trees that use the same classifier (i.e. a feature and threshold) for all the nodes that have the same depth. This means that for an oblivious decision tree of depth d, only d classifiers are learned. Said otherwise, an oblivious decision tree is a classifier that split the data using d splitting features, giving a decision table of size 2^d. To make oblivious decision trees differentiable, the authors propose to learn linear classifiers using all the features, but add a sparsity inducing operator on the weights of the classifiers (the entmax transformation). Similarly, the step function used to split the data is replaced by a continuous version (here a binary entmax transformation). Finally, the decision function is obtained by taking the outer product of all the scores of the classifiers: [c_1(x), 1-c_1(x)] o [c_2(x), 1-c_2(x)] ... This "choice" operator transforms the d dimensional vectors of the classifier scores to a 2^d dimensional vector. Another interpretation of the proposed "differentiable oblivious decision trees" is a two layer neural network, with sparsity on the weights of the first layer,
and an activation function combining the entmax transformation and the outer product operator. The authors then propose to combine multiple differentiable decision trees in one layer, giving the neural decision oblivious ensemble (NODE). Finally, several NODE layers can be combined in a dense net fashion, to obtain a deep decision tree model. The proposed method is evaluated on 6 datasets (half classification, half regression), and compared to existing decision tree methods such as XGBoost or CatBoost, as well as feed forward neural networks.

The paper is clearly written, ideas are well presented, and it is easy to follow the derivation of the method. As a minor comment, I would suggest to the authors to give more details on the EntMax method, as it is quite important for the method, but not really introduced in the paper. The proposed algorithm is sound, and a nice way to make decision trees differentiable. One concern that I have though, is that it seems that NODE are close to fully connected neural networks, with sparsity on the weights. Indeed, I think that there are two ingredients in the paper to derive the method: adding sparsity to the weights and the outer product operator (as described in the previous paragraph). In particular, the improvement over vanilla feed forward neural networks seem small in the experimental section. I thus believe that it would be interesting to study if both two differences with feed forward networks are important, or if only is enough to get better results.

To conclude, I believe that this is a well written paper, proposing a differentiable version of decision trees which is interesting. However, the proposed method relies on existing techniques, such as EntMax, and I wonder if the (relatively small) improvement compared to feed forward network comes from these. I believe that it would thus be interesting to compare the method with feed forward network with sparsity on the weights. For now, I am putting a weak reject decision, but I am willing to reconsider my rating based on the author response.

Questions to the authors:
(1) do you use the same data preprocessing for all methods (quantile transform)?
(2) would it make sense to evaluate the effects of each the entmax and the outer product operator separately in the context of fully connected networks?


**Experience Assessment:**

I do not know much about this area.

**Review Assessment: Checking Correctness Of Derivations And Theory:**

I assessed the sensibility of the derivations and theory.

**Review Assessment: Checking Correctness Of Experiments:**

I assessed the sensibility of the experiments.

**Review Assessment: Thoroughness In Paper Reading:**

I read the paper thoroughly.

---

> ### Author Response · Authors · 2019-11-09
> **R#3: We try to address your concerns.**
>
> Thank you for the insightful review. We attempt to address your comments below.
>
> [do you use the same data preprocessing for all methods (quantile transform)?]
>
> Yes, exactly. We apply the same preprocessing steps for all methods to minimize the effect of built-in preprocessing of GBDT methods like Catboost. While important, such preprocessing steps have been applied to all models including FCNN and NODE for a fair comparison.
>
> [would it make sense to evaluate the effects of each the entmax and the outer product operator separately in the context of fully connected networks?]
>
> We agree that such an investigation would be interesting. We have evaluated the oblivious decision trees without sparsity in the “softmax” column of Table 3 in the original submission. As for sparse weights without ODTs, we conduct such an experiment and report results below.
>
> For these experiments we consider three ways to sparsify weight matrices for fully-connected neural networks: row-wise Entmax (α=1.5), column-wise Entmax (α=1.5) and $L_0$ regularization[1]. We tune each setup using standard FCNN tuning procedure from the original submission.
>
> | Method   | YearPrediction |   Epsilon   |
> |--------------|---------------------|---------------|
> | FCNN       |    79.99              |   0.1041    |
> | L_0 reg    |    80.54              |   0.1132    |
> | Row-wise|    84.87              |   0.16460   |
> | Col-wise  |    81.13              |   0.16192   |
>
> Unfortunately, neither of the proposed methods was able to surpass the dense FCNN performance. Our hypothesis is that the sparsity benefits the NODE performance since it learns sparse *choice* functions. Conversely, FCNN sparse weight matrices by themselves do not improve the model’s performance on tabular data.
>
> [give more details on the EntMax method]
>
> We have added the brief description of entmax in a new revision.
>
> [1] Louizos, Christos, Max Welling and Diederik P. Kingma. “Learning Sparse Neural Networks through L0 Regularization.” ICLR 2018

---

### Decision · Program_Chairs · 2019-12-19

**Decision:**

Accept (Poster)

**Comment:**

This paper proposes Neural Oblivious Decision Ensembles, a formulation of ensembles of decision trees that is end-to-end differentiable and can use multi-layer representation learning. The reviewers are in agreement that this is a novel and useful tool, although there was some mild concern about the extent of the improvement over other methods. Post-discussion, I am recommending the paper be accepted.